# Linking Dysregulated AMPK Signaling and ER Stress in Ethanol-Induced Liver Injury in Hepatic Alcohol Dehydrogenase Deficient Deer Mice

**DOI:** 10.3390/biom9100560

**Published:** 2019-10-02

**Authors:** Mukund P. Srinivasan, Kamlesh K. Bhopale, Samir M. Amer, Jie Wan, Lata Kaphalia, Ghulam S. Ansari, Bhupendra S. Kaphalia

**Affiliations:** 1Department of Pathology, The University of Texas Medical Branch, Galveston, TX 77555, USA; 2Department of Forensic Medicine and Clinical Toxicology, Tanta University, Tanta 31512, Egypt; 3Division of Pulmonary, Critical Care Medicine, Department of Internal Medicine, The University of Texas Medical Branch, Galveston, TX 77555, USA

**Keywords:** alcoholic liver disease, alcohol dehydrogenase, AMPK signaling, ER stress, deer mice

## Abstract

Ethanol (EtOH) metabolism itself can be a predisposing factor for initiation of alcoholic liver disease (ALD). Therefore, a dose dependent study to evaluate liver injury was conducted in hepatic alcohol dehydrogenase (ADH) deficient (ADH^−^) and ADH normal (ADH^+^) deer mice fed 1%, 2% or 3.5% EtOH in the liquid diet daily for 2 months. Blood alcohol concentration (BAC), liver injury marker (alanine amino transferase (ALT)), hepatic lipids and cytochrome P450 2E1 (CYP2E1) activity were measured. Liver histology, endoplasmic reticulum (ER) stress, AMP-activated protein kinase (AMPK) signaling and cell death proteins were evaluated. Significantly increased BAC, plasma ALT, hepatic lipids and steatosis were found only in ADH^−^ deer mice fed 3.5% EtOH. Further, a significant ER stress and increased un-spliced X-box binding protein 1 were evident only in ADH^−^ deer mice fed 3.5% EtOH. Both strains fed 3.5% EtOH showed deactivation of AMPK, but increased acetyl Co-A carboxylase 1 and decreased carnitine palmitoyltransferase 1A favoring lipogenesis were found only in ADH^−^ deer mice fed 3.5% EtOH. Therefore, irrespective of CYP2E1 overexpression; EtOH dose and hepatic ADH deficiency contribute to EtOH-induced steatosis and liver injury, suggesting a linkage between ER stress, dysregulated hepatic lipid metabolism and AMPK signaling.

## 1. Introduction

Chronic alcohol consumption is one of the major causes for inflammatory liver disease, initiating from fatty liver or hepatic steatosis and progressing to fibrosis, then cirrhosis, and may lead to hepatocellular carcinoma and liver failure. About 30% of heavy drinkers with fatty liver may also develop hepatitis, and ~4% of global mortality is related to regular heavy alcohol consumption. [1,2,3]. Thus, alcoholic liver disease (ALD) is a serious health problem and risk factor for significant morbidity and mortality. As such, chronic alcohol abuse is a huge socioeconomic burden costing ~$250 billion to the U.S. economy and nearly 100,000 deaths annually [4]. Other than chronic consumption of alcohol, several potential predisposing conditions including age, gender, genetic background, nutritional status, occupational hazards and viral infection (especially HCV infection) have been implicated in the development of liver disease [5]. Evidences also indicate the role of endotoxins in alcoholic hepatitis and fibrosis [6,7,8]. Additional factors, such as epigenetics and the environment, influence pathogenesis of ALD [9,10,11]. Therefore, the mechanism(s) and metabolic basis of ethanol (EtOH)-induced fatty liver progressing to inflammation and advanced forms of ALD, such as fibrosis and cirrhosis, are not well understood.

The severity of liver disease correlates well with the amount and duration of alcohol consumption [12,13]. Over 90% of ingested EtOH is predominantly metabolized by hepatic alcohol dehydrogenase (ADH, a major enzyme involved in the oxidative metabolism of alcohol), which is also known to be impaired in humans and in experimental animals after chronic alcohol/EtOH consumption [14,15,16,17]. The pathogenesis of ALD is thought to be associated with oxidative stress due to CYP2E1-catalyzed oxidative metabolism of EtOH and/or the generation of reactive oxygen species [18]. However, the role of hepatic ADH deficiency in the initiation and progression of ALD is scarcely studied.

Chronic alcohol consumption dysregulates lipid metabolism, impairs lipid transport from the liver and causes hepatic steatosis by inducing lipogenesis and reducing the catabolism of fatty acids by beta-oxidation [19]. Typically, an accumulation of lipid droplets (LDs) in the hepatocytes in the form of triglycerides and cholesterol/cholesterol esters is an initial stage in alcoholic fatty liver. The hepatic lipogenic pathway is activated after the consumption of a mere 24 g of EtOH per day in humans [20]. Lipids derived from the diet and/or from adipose tissue by EtOH-induced activation of lipases catalyzing the breakdown of conjugated lipids, may also contribute to hepatic steatosis [21]. More recently, decreased LD degradation/lipophagy has been shown to be one of the possible factors for alcohol induced hepatic steatosis [22,23]. While the mechanism of lipid accumulation in the liver is still elusive; oxidative stress, cell death, endoplasmic reticulum (ER) stress and AMP activated kinase (AMPK) deactivation could be a common finding in murine models of ALD [24,25]. Generally, AMPK activation by drugs or chemicals prevents the pathologic progression of non-alcoholic fatty liver disease to non-alcoholic steatohepatitis by inhibiting oxidative stress and inflammation [26]. Therefore, the metabolism of EtOH can itself be one of the important contributing factors in the etiology of ALD [27]. However, interrelationships among various mechanisms involved in ALD and liver injury are also not well defined. Thus, the focus of this study was to evaluate dose-dependent chronic effects of EtOH on lipid contents, AMPK signaling and ER stress, and its responses in hepatic ADH^−^ deer mice.

## 2. Materials and Methods

### 2.1. Antibodies and Reagents

The primary antibodies for AMPKα (62 kDa; catalogue number 5831), phospho (p)-AMPKα (Thr 172) (62 kDa; catalogue number 2535), Ca^2+^/calmodulin-dependent protein kinase kinase β (CaMKKβ; 60, 50 kDa; catalogue number 4436), p-CaMKKβ (Thr286) (60, 50 kDa; catalogue number 12716), liver kinase B1 (LKB1; 54 kDa; catalogue number 3050), p-LKB1 (Ser 428) (54 kDa; catalogue number 53482), acetyl CoA carboxylase 1 (ACC1; 265 kDa; catalogue number 4190), p-ACC1 (Ser 79) (280 kDa; catalogue number3661), fatty acid synthase (FAS; 273 kDa; catalogue number 3189), glucose regulated protein 78 (GRP78; 78 kDa; catalogue number 3117), eukaryotic translation initiation factor 2α (eIF2α; 38 kDa; catalogue number 5324), p-eIF2α (Ser 51; 38 kDa; catalogue number 3398), caspase-3 (17, 19, 35 kDa; catalogue number 4436), caspase-8 (10, 57 kDa; catalogue number 4790) and β-actin (45 kDa; catalogue number 4970) were purchased from Cell Signaling Technology (Danvers, MA). Antibodies for spliced X-box binding protein 1 (sXBP1; 40kDa; catalogue number 37152), unspliced XBP1 (uXBP1; 29 kDa; catalogue number 37152), carnitine palmitoyltransferase 1A (CPT1A; 88 kDa; catalogue number 128568), 4-hydroxynonenal (4-HNE; catalogue number ab46545), CD3 (catalogue number ab5690) and caspase-1 (45 kDa, catalogue number 1872) were purchased from Abcam Inc (Cambridge, MA, USA). Antibodies to inositol-requiring transmembrane kinase/endoribonuclease 1α (IRE1α; 110 kDa; catalogue number NB100-2324), p-IRE1α (Ser 724) (110 kDa; catalogue number NB100-232), activating transcription factor 6 (ATF6; 88 kDa; catalogue number IMG273) and sterol regulatory element-binding protein 1 (SREBP1c; 120 kDa; catalogue number NB600-582) were from Novus Biologicals (Littleton, CO, USA). Antibodies to protein kinase R-like endoplasmic reticulum kinase (PERK; 150 kDa; catalogue number 100-401-962) and CHOP (31 kDa; catalogue number MA1-250) were from Rockland (Limerick, PA, USA) and Thermo Fisher Scientific (Houston, TX, USA) respectively. P-nitrophenol (purity >99% pure), and other chemicals (reagents with HPLC and GC grade solvents) used in the present study were obtained from Sigma Aldrich Co. (St Louis, MO, USA) or Thermo Fisher Scientific (Houston, TX, USA).

### 2.2. Animal Experiments

One-year old male hepatic class 1 ADH negative phenotype (ADH^−^) deer mice, a natural genetic variant of *Peromyscus maniculatus*, and ADH positive (ADH^+^) deer mice obtained from *Peromyscus* Stock Center, University of South Carolina, Columbia, SC, were housed in UTMB’s Animal Resource Center. Both strains were divided into two groups, control and experimental. All the animal experiments conducted in this study were in accordance with animal care protocols instituted by UTMB’s Institutional Animal Care and Use Committee. Experimental groups were fed a Lieber–DeCarli liquid diet (Dyets Inc., Bethlehem, PA, USA) for a week, followed by EtOH in the liquid diet, maintained at 1%, 2% or 3.5% (*w*/*v*) daily for two months [28,29]. Controls for both strains were pair-fed with liquid diets containing EtOH equivalent calories substituted by maltose-dextrin. Animals were anesthetized by intraperitoneal administration of Nembutal^®^ (Sodium salt, 80 mg/mL) at the end of 2 months of EtOH feeding and blood was collected by cardiac puncture in the heparinized tubes. Fifty microliters of whole blood was transferred to gas chromatography vials for analyzing blood alcohol and acetaldehyde concentrations, and the remaining blood was centrifuged at 1000× *g* for 10 min. Plasma was separated and stored at −80 °C. Livers were excised for gross examination and a portion was fixed in 10% buffered formalin, dehydrated in 70% EtOH and embedded in paraffin blocks for histology and immunohistochemistry [27,28]. For electron microscopic examination, sections of livers were fixed in a buffered mixture of formaldehyde and glutaraldehyde and embedded in epoxy plastic [27]. Remaining portions of the livers were stored at −80 °C for biochemical and molecular studies.

### 2.3. Blood Alcohol and Acetaldehyde Levels, Plasma ALT and Hepatic CYP2E1 Activity

Blood alcohol and acetaldehyde levels were analyzed by using headspace gas chromatography (GC) as described previously [29]. A key marker of liver injury, alanine aminotransferase (ALT) was assayed in the plasma using kit from Biotron Diagnostics Inc (Hemet, CA, USA) as per the manufacturer’s instructions. The hepatic CYP2E1 activity was determined by measuring the rate of oxidation of p-nitrophenol (PNP) to p-nitrocatechol in the presence of NADPH and O_2_ [30]. In brief, 100-μL reaction mixture consisted of 200 μg of liver homogenate in 100 mM potassium-phosphate buffer (pH 7.4), with 0.2 mM PNP. The reaction was initiated by adding 1 mM NADPH at 37 °C and interrupted at 60 min by an addition of 30 μL of 20% trichloroacetic acid, and the supernatant was treated with 10 μL of 10 M sodium hydroxide. The absorbance was monitored at 546 nm and the activity determined using an equation—OD546/9.53/0.2/60/7.1 × 106. The CYP2E1 activity was expressed as pmol/min/mg of protein.

### 2.4. Morphological Studies and Immunohistochemical Staining

Standard sections of fixed liver tissue embedded in paraffin blocks were cut (5 μm thick) and stained with Hematoxylin and Eosin (H&E) for the light microscopic examination [27]. Thin liver tissue sections were processed for the immunohistochemical detection of oxidative stress and inflammation using antibodies against 4-hydroxynonenal (4-HNE) and CD3, respectively [28,31]. Thin liver tissue sections were also processed for immunohistochemical detection of endoplasmic reticulum (ER) stress using an antibody against GRP78. For electron microscopy, ultrathin sections were cut and examined with a Philips 201 or CM-100 electron microscope, as described previously [27].

### 2.5. Hepatic Lipid Contents

Frozen liver tissue (250 mg) was homogenized and the lipids were extracted as described previously [32]. Quantitative determination of hepatic triglycerides, total cholesterol, esterified cholesterol and non-esterified fatty acids (NEFA) were done using respective assay kits from Wako diagnostics (Richmond, VA, USA) according to manufacturer’s instructions.

### 2.6. Immunoblot Analysis

AMPKα and its upstream and downstream signaling, ER stress and related unfolded protein response (UPR) signaling molecules and apoptosis related cell death proteins were analyzed by western blot in the hepatic post nuclear fraction. In brief, protein content of the post nuclear fraction was measured by Bradford protein assay kit (Bio-Rad, Hercules, CA, USA). A 30 μg protein aliquot from each fraction was electrophoresed using precast 4%–12% NuPage mini-gels (Life Technologies, Carlsbad, CA, USA), and the resolved proteins were transferred onto PVDF membrane (EMD Millipore), blocked with 5% milk in Tris-buffered saline with tween and probed with their respective primary antibodies (1:1000 dilution). After extensive washes, the immunoreactivity was detected using specific horseradish peroxidase-conjugated secondary antibodies (1:5000 dilution), followed by enhanced chemiluminescence, as described [28]. The protein bands were quantified using NIH Image J Software (version 1.50i, Bethesda, PA, USA) and normalized to loading control β-actin values. The ratio of phosphorylated protein to total protein was then calculated.

### 2.7. Statistical Analysis

Data are expressed as means ± SEMs (standard errors of means) of ≥5 animals per group unless otherwise indicated. The data sets were analyzed for statistical significance using Student’s *t*-tests and one-way ANOVA followed by Tukey’s multiple comparison test. A *p*-value ≤ 0.05 was considered significant.

## 3. Results

Overall, predominant hepatic steatosis and significant ER stress, with an increased lipid accumulation were observed only in the livers of ADH^−^ compared to ADH^+^ deer mice fed 3.5% EtOH. Although no significant changes in cell death pathways and oxidative stress were indicated in both EtOH-fed ADH^−^ and ADH^+^ deer mice groups, changes in injury marker, histology and ER stress found in the ADH^−^ compared to ADH^+^ deer mice fed 3.5% EtOH could be attributed to total body burden of EtOH and/or its metabolism irrespective of hepatic CYP2E1 expression. Although AMPKα deactivation was observed in both ADH^−^ and ADH^+^ deer mice fed 3.5% EtOH, a differential expression of CaMKKβ, ACC1 and CPT1A found in ADH^−^ deer mice fed 3.5% EtOH is particularly important to understand the metabolic basis of EtOH-induced steatosis and liver injury.

### 3.1. Morphological and Immunohistochemical Changes

At the microscopic level, significant hepatic steatosis was found only in ADH^−^ versus ADH^+^ deer mice fed 3.5% EtOH using H&E staining [Figure 1A (f)]. Similar changes were not observed in both strains fed 1% or 2% EtOH. Significant oxidative stress and inflammatory response were not observed in the liver sections of both strains at all doses of EtOH used in this study (data not shown). Immunohistochemistry using GRP78 specific antibody showed a significant ER stress only in livers of ADH^−^ deer mice fed 3.5% EtOH [Figure 1B(f)]. Quantification of immunohistochemical staining for GRP78 in the livers of ADH^−^ compared to ADH^+^ deer mice fed 3.5% EtOH is shown in Figure 1C. In addition, electron micrographs of the liver sections showed dilation and swelling of endoplasmic reticulum (ER) and ER cisternae only in ADH^−^ compared to ADH^+^ deer mice fed 3.5% EtOH (Figure 2A,B). Thus, hepatic ADH deficiency and the dose of EtOH together could be determining factors in EtOH-induced hepatic steatosis and ER stress.

### 3.2. Blood Alcohol and Acetaldehyde Levels, and Levels of Hepatic Injury Marker and CYP2E1

Dose-dependent increases for blood alcohol concentration (BAC; Figure 3A) were observed for EtOH-fed ADH^−^ versus ADH^+^ deer mice. BAC was <10 mg/dL or mg% in the deer mice fed 1% and 2% EtOH, but exponentially increased in both strains fed 3.5% EtOH. The average BAC was about two-fold greater in ADH^−^ deer mice (~148 mg%) compared to ADH^+^ deer mice (~78 mg%) fed 3.5% EtOH. Blood acetaldehyde was also increased in ADH^−^ deer mice fed 3.5% EtOH, but the average concentrations were not significantly different than those of its pair-fed control (Figure 3B). In addition, plasma ALT activity, a marker for hepatic injury, was significantly elevated (about two-fold) only in ADH^−^ deer mice fed 3.5% EtOH compared to its pair-fed control (Figure 3C). However, similar changes in ADH^+^ deer mice fed EtOH compared to its pair-fed controls were not apparent.

Hepatic CYP2E1 activity, as implicated in EtOH-induced oxidative stress in ALD, was significantly increased in both strains fed 1% or 2% EtOH, but the activity was significantly increased only in ADH^+^ deer mice fed 3.5% EtOH (Figure 3D). A similar change for hepatic CYP2E1 was not found for ADH^−^ deer mice fed 3.5% EtOH versus its pair-fed control. This is a paradox, because there was no steatosis and liver injury despite dose-dependent increases for CYP2E1 activity in EtOH-fed ADH^+^ deer mice. Interestingly, significant oxidative stress was also not found in either strains at all doses of EtOH used in the present study (data not shown).

### 3.3. Hepatic Lipid Levels

Significant increases in hepatic triglycerides (Figure 4A), NEFA (Figure 4B) and esterified cholesterol (Figure 4C) by ~1.5, 1.4 and 2.2-fold, respectively, were observed only in ADH^−^ deer mice fed 3.5% EtOH compared to the pair-fed controls. Further, a significant dose-dependent increase for total cholesterol (Figure 4D) was found in ADH^−^ deer mice fed 2% and 3.5% EtOH compared to their respective pair-fed controls. However, similar changes were not found in ADH^+^ deer mice fed 1%, 2% or 3.5% EtOH. These results indicate that hepatic ADH deficiency contributes and alters hepatic lipid metabolomics only at a 3.5% EtOH dose.

### 3.4. ER Stress and Cell Death Pathways

As mentioned earlier, a significant increase for hepatic GRP78 expression was observed in ADH^−^ versus ADH^+^ deer mice fed 3.5% EtOH, as assessed by the immunostaining (Figure 1B,C). Surprisingly, key ER stress responses (UPR regulators), and the expression of cell death proteins (caspase-1, caspase-3 and caspase-8) determined in livers of ADH^−^ and ADH^+^ deer mice fed 1%, 2% or 3.5% EtOH remained unchanged (Table 1), except the significantly increased expression of uXBP1 only in ADH^−^ deer mice fed 3.5% EtOH (Figure 5A). The uXBP1 data in ADH^−^ deer mice fed 3.5% EtOH suggests a splicing block for XBP1, a key transcriptional and translational molecule required for the operational, adaptive mechanism(s) of ER homeostasis. A significant change was also not observed for protein levels of p-IRE1α, sXBP1, ATF-6, p-EIF2α, p-PERK and CHOP in ADH^−^ and ADH^+^ deer mice fed 1%, 2 % or 3.5% EtOH, respectively (Table 1).

### 3.5. AMPKα Signaling

As shown in Figure 6A, a significant decrease in AMPKα phosphorylation was observed in both ADH^−^ and ADH^+^ deer mice fed 3.5% EtOH compared to their respective pair-fed controls. There was no significant change for phosphorylation of AMPKα in ADH^−^ and ADH^+^ deer mice fed 1% or 2% EtOH respectively. However, both an increased expression of ACC1 (a key protein involved in lipogenesis) (Figure 6B) and decreased expression of CPT1A (a key protein involved in β-oxidation of fatty acids) (Figure 6C), downstream of AMPKα signaling cascade were seen only in ADH^−^ deer mice fed 3.5 % EtOH compared to pair-fed controls. These are the key findings. An increased expression of FAS (protein involved in fatty acid synthesis) (Figure 6D) was observed in both strains fed 2% EtOH compared to pair-fed controls, indicating that FAS may not be contributing significantly in the process of EtOH-induced steatosis, since steatosis was not observed in ADH^+^ deer mice fed EtOH. Similarly, SREBP1 expression was not altered in either strain of deer mice at all doses of EtOH. The expression of various proteins involved in AMKP signaling, and SREBP1, in the livers of ADH^−^ deer mice fed 3.5% EtOH, are summarized in Table 2.

A significant decrease for CaMKK-β phosphorylation, an upstream signaling molecule of AMPKα, was observed only in ADH^−^ deer mice fed 3.5% EtOH compared to pair-fed control and EtOH-fed ADH^+^ deer mice (Figure 6E). These findings indicate the significance of Ca^2+^ metabolism and ER stress in cellular energetics involving AMPKα signaling. However, LKB1, an upstream regulator of AMPKα associated with oxidative stress, was not changed in either ADH^−^ or ADH^+^ deer mice fed 1%, 2% or 3.5% EtOH. In our animal model, LKB1 expression levels indicate that oxidative stress may not be involved in EtOH-induced liver injury, which is also supported by immunohistochemical findings using 4-HNE antibodies (data not shown). AMPKα inactivation could be linked to EtOH induced ER stress in addition to increased lipid accumulation, in part due to a significantly decreased p-ACC1/ACC1 ratio and CPT1A in ADH^−^ versus ADH^+^ deer mice. However, the significance of aforementioned changes between ADH^−^ and ADH^+^ deer mice regarding AMPKα downstream signaling and their interrelationship with steatosis needs further investigation.

## 4. Discussion

About 50% of all the deaths due to liver disease/cirrhosis worldwide are associated with chronic alcohol consumption [33]. However, no effective treatment is available for alcoholic liver disease (ALD) due to a lack of clear understanding regarding its mechanism. Chronic alcohol consumption is associated with fatty liver (an early reversible form of the disease), followed then by hepatitis (necrotic and inflammatory), fibrosis and cirrhosis—where normal hepatic parenchyma is replaced by thick bands of fibrous tissue and regenerative nodules, resulting in such clinical manifestations as portal hypertension and liver failure. Initial events, such as a dysregulated metabolism of lipids and impaired lipid transport from the liver are thought to be involved in fatty liver formation resolving into ALD. However, progression of ALD appears to be an interplay of various underlying processes, such as ER stress, the deactivation of liver-specific AMPK, oxidative stress and mitochondrial toxicity.

The activation of liver specific AMPK has been shown to protect against hepatic triglyceride accumulation, a hallmark of non-alcoholic fatty liver disease [34]. Therefore, AMPK deactivation by EtOH, in the ADH^−^ deer mouse model, appears to be one of the factors leading to the accumulation of lipids in the liver. Thus, the mechanism of ALD may also include the metabolic effects of EtOH on multiple pathways and biochemical events, including the pre-existing status of alcohol metabolizing enzymes in the target organs.

Hepatic class I ADH (an isoform responsible for majority of oxidative metabolism of EtOH) deficiency appears to be determining factor for exponentially elevated BAC in ADH^−^ versus ADH^+^ deer mice fed 3.5% EtOH. An increased blood EtOH elimination rate (BEER) reported in ADH^+^ deer mice after chronic EtOH consumption is believed to be mediated primarily by hepatic class I ADH catalyzed EtOH metabolism, and to a lesser extent, via induction of the hepatic microsomal EtOH-oxidizing system (CYP2E1) [35]. Furthermore, class I ADH is shown to increase EtOH elimination rate and decrease the EtOH concentration curve [36]. Overall, high BAC in ADH^−^ deer mice fed 3.5% EtOH could be due to hepatic ADH deficiency despite significant CYP2E1 induction, as also supported by the studies in ADH1 knockout mice [37]. Relatively low BAC in ADH^+^ mice fed EtOH could be attributed to increased BEER [35]. An increasing trend of BAC with an increasing dose of EtOH, as found in both strains, shows dose dependency, but an about two-fold greater BAC and liver injury with significant steatosis in ADH^−^ versus ADH^+^ deer mice fed 3.5% EtOH, suggests a significant role of hepatic class I ADH in EtOH-induced liver injury. In contrast, no significant changes for blood acetaldehyde concentration between the control and EtOH-fed groups, despite increasing doses of EtOH suggests that acetaldehyde did not play a significant role in EtOH induced liver damage in our model. In addition, it also highlights a lack of its specificity as a marker of EtOH intake/liver injury compared to BAC. The formation of acetaldehyde catalyzed by ADH and/or CYP2E1 is implicated in EtOH-induced oxidative stress [18]. However, an induction of hepatic CYP2E1 in both strains fed EtOH, irrespective of liver injury, indicates that CYP2E1 may not be a major contributing factor in EtOH-induced liver injury in the deer mouse model at doses used in this study. A lack of significant oxidative stress, as observed by immunostaining in the livers of EtOH-fed mice of both strains, and increased levels of CYP2E1 in ADH^+^ versus ADH^−^ deer mice fed 3.5% EtOH, further rules out a significant contribution of CYP2E1 in EtOH metabolism related liver injury.

Dysregulated metabolism of lipids (increased hepatic triglycerides, NEFA, esterified and total cholesterol) appears to be associated with significantly greater steatosis, as found in the livers of ADH^−^ deer mice fed 3.5% EtOH. These findings are parallel to our previously published NMR-based lipidomic studies in the livers of both strains fed EtOH [31]. Fatty acids, through diet, lipolysis of triglycerides in adipose tissue and/or impaired oxidation of fatty acids can accumulate lipids in the liver [20,38]. Lipids may also accumulate in the liver through increased de novo synthesis and/or their impaired export [39,40]. However, upregulation of FAS in the livers of both strains fed EtOH, as found in this study seems unrelated to formation of fatty liver.

Steatosis is an early step in ALD, which could be a risk factor for such pathologies as hepatitis, cirrhosis and even hepatocellular carcinoma. A greater extent of steatosis appears to be associated with liver injury with increased levels of plasma ALT, triglycerides, NEFA, and esterified and total cholesterol, as found only in 3.5% EtOH-fed ADH^−^ deer mice. However, dose dependent increases for total cholesterol, as found in ADH^−^ deer mice fed EtOH, could also contribute to development of fatty liver. Since SREPB1c remained unaltered in both strains irrespective of EtOH doses used in this study, alternative mechanism(s) for increased total cholesterol in ADH^−^ deer mice fed EtOH remains to be investigated. Nevertheless, an intervention of the disease at an early stage could save huge health care costs associated with the treatment of advanced stages of ALD.

A significant endoplasmic reticulum (ER) stress evident by increased levels of GRP78, as demonstrated by immunostaining in ADH^−^ deer mice fed 3.5% EtOH, could be associated with steatosis and liver injury. The unfolded protein response (UPR) regulates and restores protein folding. While anorexia-induced lipolysis promotes late triglyceride and free fatty acid accumulation, an impaired fatty acid oxidation can also contribute to an early development of steatosis in the liver. Such findings also provide evidence for both direct and indirect regulation of peripheral metabolism by ER stress [41]. Chronic alcohol consumption induces ER stress and disrupts cellular protein homeostasis and proliferation, and cell cycle progression, promoting the development of advanced stages of liver diseases, including hepatocellular carcinogenesis [42]. ER stress responses play critical roles in maintaining protein homeostasis in the secretory pathway to avoid damage to the host through three, well-orchestrated pathways collectively known as UPR. Importantly, inflammation induced by ER stress can also directly be responsible for the pathogenesis of metabolic and inflammatory diseases. A prolonged ER stress has long been implicated in a variety of diseases in metabolic, inflammatory and malignant conditions [43]. Increased GRP78 (ER stress sensor), despite no significant changes for IRE1α and ATF 6 (two of the three UPR transducers), and significant increases for uXBP1 only in the livers of ADH^−^ deer mice fed 3.5% EtOH, could be linked to prolonged ER stress. Thus, an accumulation of uXBP1 is self-explained, since spliced XBP1 is an important translational and transcriptional regulator involved in homeostasis of ER membrane. No significant changes observed caspase-1, caspase-3, caspase-8 (cell death proteins) in our study further suggest an early stage of liver injury and a lack of apoptosis in our model.

AMPKα, central regulator/sensor of cellular energy homeostasis, has been considered an important therapeutic target for controlling human diseases, including metabolic syndrome, diabetes and cancer [44]. AMPKα is a serine/threonine protein kinase complex consisting of a catalytic α-subunit. A variety of conditions that deplete cellular energy levels, such as nutrient starvation, hypoxia and exercise can activate AMPK. Activity of AMPK is also regulated by upstream kinases, LKB 1 and CaMKKβ, related to oxidative stress and ER stress, respectively. Inhibition of LKB1 and/or CaMKKβ could also deactivate AMPK [45]. Since AMPK activation plays a key role in maintaining the balance between anabolic and catabolic pathways for cellular homeostasis in response to metabolic stress, its deactivation could be involved in upregulation of lipogenesis and down regulation of fatty acid oxidation, resulting in lipid accumulation within the cell. Although emerging evidence indicates role of AMPK deactivation in EtOH-induced steatosis and liver disease [46], a precise mechanism(s) and the cascade of events leading to ALD have not been fully understood. Several drugs, antioxidants and agents that activate AMPK are also known to reinstate cellular redox status and ER homeostasis [25,26,47]. Therefore, dysregulated AMPK signaling and ER stress could be linked with etiopathogenesis of ALD. Significantly decreased CPT1A in ADH^−^ versus ADH^+^ deer mice fed 3.5% EtOH and decreased ratios for p-AMPK/AMPK and p-ACC1/ACC1 could be key determinants for increased EtOH-induced hepatic steatosis in ADH^−^ deer mice fed 3.5% EtOH. However, other potential mechanisms for steatosis in the deer mouse model also need to be considered regarding the role of dietary lipids and the mobilization of free fatty acids from adipose tissue.

Of note, decreased lipid droplets’ (LDs) degradation/lipophagy is meant to be a major determining factor for an elevated lipid accumulation in the liver, leading to steatosis [48]. Evidence suggests, that chronic alcohol consumption retards lipophagic clearance of LDs, which contributes to alcohol induced steatosis [22,23,49]. The process of lipophagy is regulated by AMPK [50] can be dysregulated by EtOH and its metabolites [49]. Although we observed AMPK deactivation in the livers of EtOH-fed mice of both strains, a decreased blood EtOH clearance as observed only in ADH^−^ deer mice fed 3.5% EtOH suggest an inhibition of lipophagy/degradation of LDs leading to steatosis and hepatic lipid accumulation in our model. It is likely that inhibition of hepatic class I ADH, as seen in chronic alcoholics, appears to be a determining factor for alcohol induced liver injury. Further studies are warranted to determine the extent to which EtOH metabolism and cellular energetics regulate lipophagy in deer mouse models.

## 5. Conclusions

Overall, the body burden EtOH, as shown by BAC, and hepatic class 1 ADH deficiency together, appear to be important risk factors for steatosis, ER stress and altered upstream and downstream AMPKα signaling in the liver. Along with inhibition of CPT1A, increased ACC1 and other lipogenesis pathways, mobilization of lipids to the liver and decreased lipophagy may have contributed to EtOH-induced steatosis in our deer mouse model. As presented in Figure 7, our findings in this study indicate a link of dysregulated AMPK signaling, ER stress and its responses (UPR) and lipid metabolism with steatosis and liver injury; and EtOH dose as well as hepatic ADH deficiency seem to be key factors in the process.

## Figures and Tables

**Figure 1 biomolecules-09-00560-f001:**
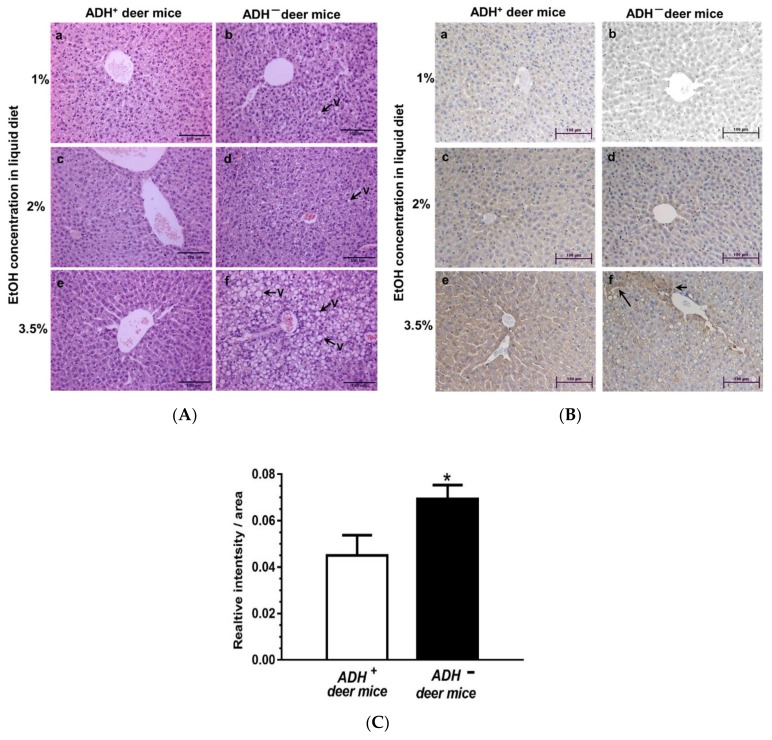
H&E (**A**) and GRP 78 immunohistochemical (IHC) (**B**) staining of representative liver tissue sections from 1%, 2% or 3.5 % EtOH-fed ADH^+^ (**a**,**c**,**e**) and ADH^−^ (**b**,**d**,**f**), respectively, fed daily for 2 months. Dose-dependent hepatic steatosis (fat vacuolization) with significant ER stress (GRP78) was distinctly observed in the liver sections of ADH^−^ deer mice fed 3.5% EtOH (original magnification × 20). The pair-fed controls of both strains showed normal histology and no positive staining for GRP78 antibodies. Quantification of the intensity of immunohistochemical staining for GRP78 (**C**) in the livers of ADH^+^ versus ADH^−^ deer mice fed 3.5% EtOH for 2 months. Data are presented as means ± standard errors of means (SEMs) (*n* = 5); * *p* < 0.05.

**Figure 2 biomolecules-09-00560-f002:**
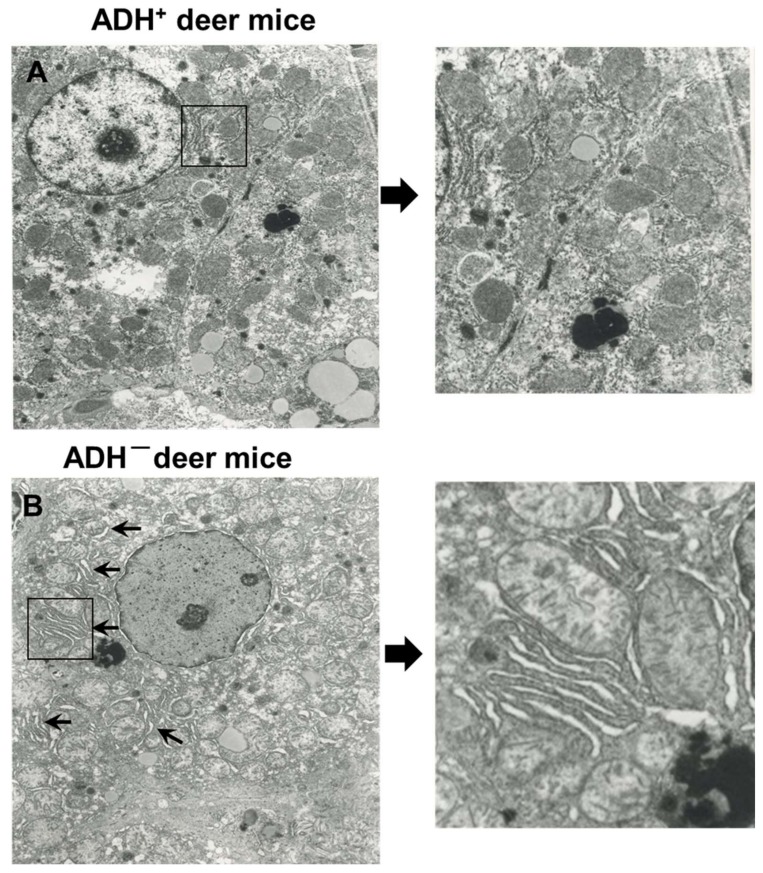
Electron micrographs of liver sections of ADH^+^ deer mice (**A**) and ADH ^–^ deer mice (**B**) fed 3.5% EtOH for 2 months. Arrows and magnified image showing dilation and swelling of ER and ER cisternae only in the hepatocytes of ADH^−^ deer mice fed 3.5% EtOH (original magnification × 14500). However, the ER appears normal in hepatocytes of ADH^+^ deer mice fed 3.5% EtOH.

**Figure 3 biomolecules-09-00560-f003:**
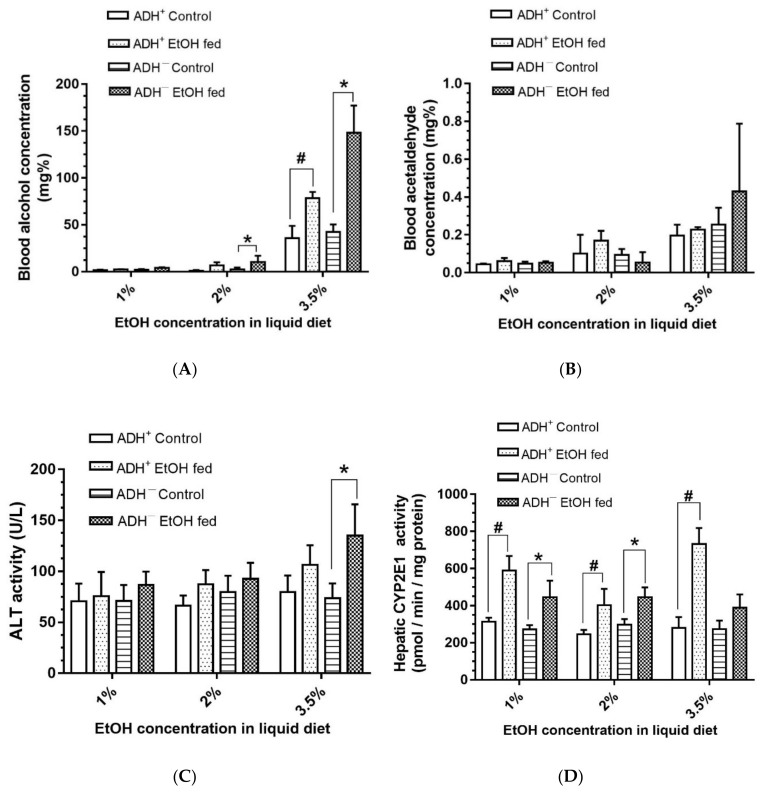
Levels of blood alcohol (**A**), blood acetaldehyde (**B**), plasma ALT activity (**C**) and hepatic CYP2E1 (**D**) in ADH^+^ and ADH^−^ deer mice fed 1%, 2% or 3.5% EtOH daily for 2 months. Values are means ± SEMs (*n* = 5). * *p* ≤ 0.05 for EtOH-fed ADH^−^ versus pair-fed control ADH^−^ deer mice. # *p* ≤ 0.05 for EtOH-fed ADH^+^ versus pair-fed ADH^+^ deer mice.

**Figure 4 biomolecules-09-00560-f004:**
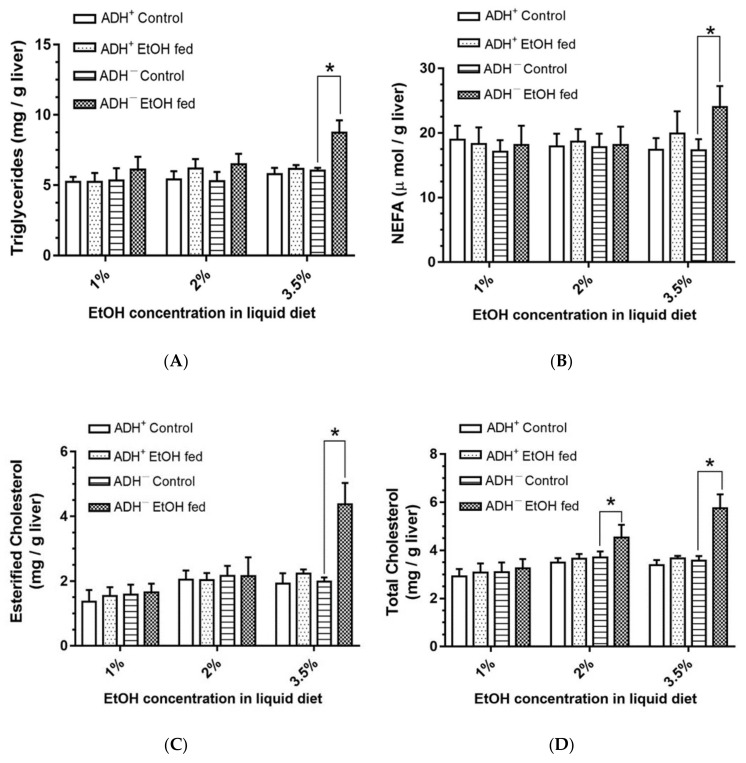
Triglycerides (**A**), non-esterified fatty acids (NEFA) (**B**), esterified cholesterol (**C**) and total cholesterol (**D**) levels in the livers of ADH^−^ and ADH^+^ deer mice fed 1%, 2% or 3.5% EtOH daily for 2 months. Values are means ± SEMs (*n* = 5). * *p* ≤ 0.05 for EtOH-fed ADH^−^ versus pair-fed control ADH^−^ deer mice.

**Figure 5 biomolecules-09-00560-f005:**
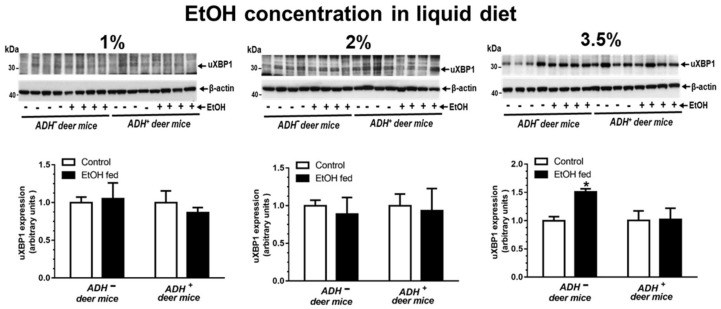
Protein expression (**upper panel**) and relative intensities (**lower panel**) for unspliced-XBP1 (uXBP1) in the livers of ADH^−^ and ADH^+^ deer mice fed 1%, 2% or 3.5% EtOH daily for 2 months. Intensities were normalized to β-actin (loading control). Values are expressed as means ± SEMs (*n* = 4). * *p* value ≤ 0.05.

**Figure 6 biomolecules-09-00560-f006:**
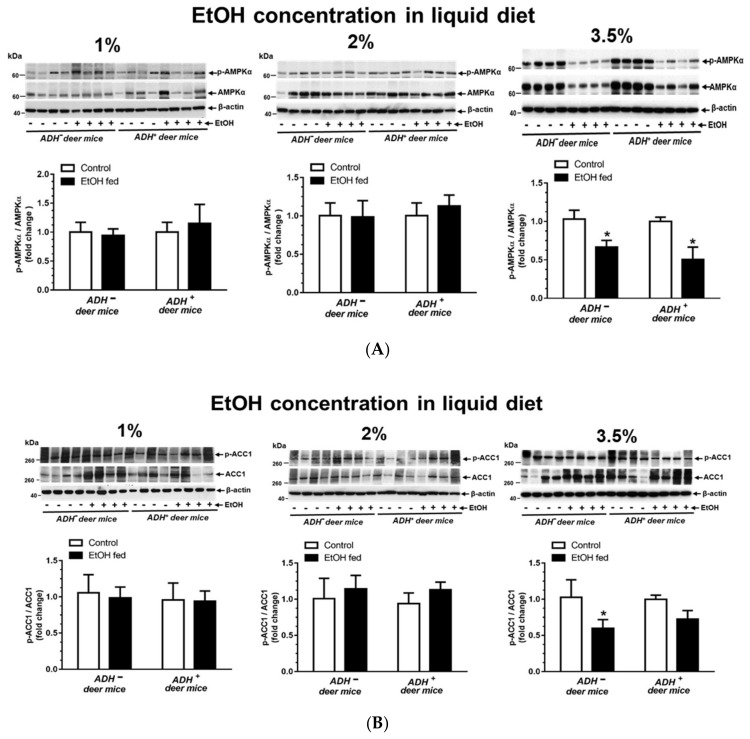
EtOH-induced altered expression for AMPKα signaling in the livers of ADH^−^ and ADH^+^ deer mice fed 1%, 2% or 3.5% EtOH daily for 2 months. Protein phosphorylation/expression for p-AMPKα/AMPKα (**A**), p-ACC1/ACC1 (**B**), CPT1A (**C**), FAS (**D**) and CaMKKβ (**E**). Western blots along with respective bar diagrams show relative intensities of p-AMPK/AMPK, p-ACC1/ACC1, CPT1A, FAS and p-CaMKKβ/CaMKKβ. Intensities were normalized to β-actin (loading control). Values are expressed as means ± SEMs (*n* = 4). * *p*-value ≤ 0.05.

**Figure 7 biomolecules-09-00560-f007:**
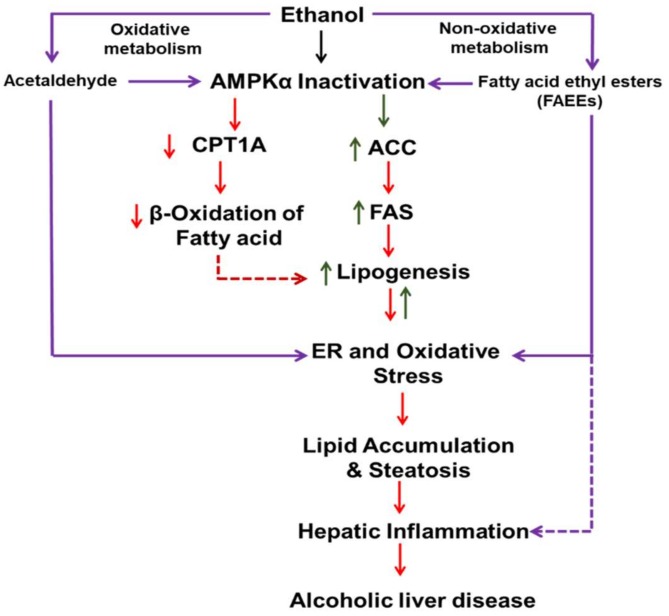
Schematic presentation showing the metabolic basis of EtOH-induced liver injury and the role of dysregulated AMPKα signaling and ER stress.

**Table 1 biomolecules-09-00560-t001:** Expression of proteins involved in ER stress and their responses, and cell death proteins in the livers of ADH^−^ deer mice fed 3.5% EtOH compared to pair-fed controls

Protein	Changes in Protein Levels/Expression
p-IRE1α/IRE1α	No Change
Unspliced XBP1 *	Increased *
ATF-6	No Change
PERK	No Change
p-EIF2α/EIF2α	No Change
CHOP	No Change
Caspase-1	No Change
Caspase-3	No Change
Caspase-8	No Change

n = 4. * *p*-value ≤ 0.05.

**Table 2 biomolecules-09-00560-t002:** Expression of proteins involved in AMPKα signaling in livers of ADH^−^ deer mice fed 3.5% EtOH versus pair-fed control.

Protein	Changes in Protein Levels/Expression
p-AMPKα/AMPKα *	Decreased *
p-ACC/ACC *	Decreased *
FAS	No Change
CPT1A *	Decreased *
SREBP1	No Change
p-CaMKKβ/CaMKKβ *	Decreased *
p-LKB1/LKB1	No Change

n = 4. *****
*p* value < 0.05.

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
