# Peer review of "Linking Dysregulated AMPK Signaling and ER Stress in Ethanol-Induced Liver Injury in Hepatic Alcohol Dehydrogenase Deficient Deer Mice"

_biomolecules, 2019, doi:10.3390/biom9100560_

Round 1

Reviewer 1 Report

In the manuscript “Linking dysregulated AMPK signaling and ER stress in ethanol-induced Liver injury in hepatic alcohol dehydrogenase deficient deer mice”, the authors conducted a dose response study in hepatic alcohol dehydrogenase deficient (ADH-) and normal (ADH+) mice fed 1, 2 and 3.5% ethanol daily for 2 months. They evaluated hepatic lipid content, liver injury (ATL levels), AMPK signaling and ER stress. They concluded that ethanol dose and hepatic ADH deficiency contribute to ethanol-induced steatosis and liver injury, and are linked with hepatic lipid dysfunction, ER stress and dysregulated AMPK signaling.

In my view, the manuscript is premature for publication for the following reasons:

The specific goal and the conclusion of the study are not clearly defined. Only ER stress marker (GRP78) was used; the immunohistochemical staining as well as the western blot data are not convincing. ALT levels are not always associated with liver injury. Only speculations about AMPK signaling pathway involvement are provided The hepatic lipid profile is interesting but no clear interpretation of the involvement of these changes in alcoholic liver disease (ALD), especially the cholesterol data. Additional mechanistic/metabolic studies are needed to depict the role of alcohol metabolism in the pathogenesis of ALD and its progression. How this study, using mice expressing or not the AHD1 isoform help in understanding the role of impairment in alcohol metabolism in human ALD where different isoforms of ADH exist?

Author Response

Responses to Reviewers' Comments:

We thank the Reviewers for the constructive comments and helpful suggestions to further improve the quality of the manuscript. As advised, we have quantified the immunohistochemical staining for GRP78 (Fig 1C). We have now included the electron microscope findings which exhibits ER stress in the livers of ADH- vs. ADH+ deer mice fed 3.5% EtOH (Fig. 2). In addition, we provided a supplementary table which shows changes in the expression of hepatic proteins of ADH- vs. ADH+ deer mice used in this study. A schematic presentation depicting pathogenies for alcoholic liver disease is provided.  We have now carefully read all the comments and revised the manuscript accordingly as outlined below: 

Reviewer #1:

Comment: In the manuscript “Linking dysregulated AMPK signaling and ER stress in ethanol-induced liver injury in hepatic alcohol dehydrogenase deficient deer mice”, the authors conducted a dose response study in hepatic alcohol dehydrogenase deficient (ADH-) and normal (ADH+) mice fed 1, 2 and 3.5% ethanol daily for 2 months. They evaluated hepatic lipid content, liver injury (ATL levels), AMPK signaling and ER stress. They concluded that ethanol dose and hepatic ADH deficiency contribute to ethanol-induced steatosis and liver injury, and are linked with hepatic lipid dysfunction, ER stress and dysregulated AMPK signaling. In my view, the manuscript is premature for publication for the following reasons:

The specific goal and the conclusion of the study are not clearly defined. Only ER stress marker (GRP78) was used; the immunohistochemical staining as well as the western blot data are not convincing. ALT levels are not always associated with liver injury. Only speculations about AMPK signaling pathway involvement are provided the hepatic lipid profile is interesting

but no clear interpretation of the involvement of these changes in alcoholic liver disease (ALD), especially the cholesterol data. Additional mechanistic/metabolic studies are needed to depict

the role of alcohol metabolism in the pathogenesis of ALD and its progression. How this study, using mice expressing or not the AHD1 isoform help in understanding the role of impairment in

alcohol metabolism in human ALD where different isoforms of ADH exist?

Response: As pointed out by the reviewer, we have clarified our specific goals and conclusions of the study in the revised manuscript. As pointed out by the reviewer, we have now quantified the IHC staining of GRP78 (Fig 1C), using NIH Image J Software (version 1.50i, Bethesda, NIH) [1]. The quantification of IHC showed a significant difference in GRP78 staining between ADH- vs ADH+ deer mice fed 3.5% EtOH. In addition, electron microscopic images of liver sections showing ER stress only in ADH- deer mice fed 3.5% EtOH supports the ER stress findings (Fig 2). However, we did not observe changes in other ER stress markers, except a significantly increased un-spliced (u) XBP1. A significant increase for uXBP1 in ADH- vs. ADH+ deer mice fed 3.5% EtOH suggests a lack of IRE1α induction as found in this study (Table 1). An increased level of uXBP1 and a lack of increased levels of spliced XBP1 (an active transcription factor, Table 1) are suggestive of failure of activation of unfolded protein response (UPR) to reinstate the ER homeostasis. Increased lipid levels and steatosis observed in the livers of ADH- deer mice fed 3.5% EtOH suggests ER stress has occurred, as ER stress itself can also lead to lipid accumulation, or hepatic steatosis indicating that ER stress regulates, directly or indirectly, lipid metabolism in liver [2]. Further, increasing the duration of ethanol feeding beyond 2 months could provide basis for ER stress in the pathogenesis of ALD in our model.

Serum ALT levels are broadly used as a sensitive indicator of liver injury [3].  ALT is a more specific indicator of liver injury because of its cytosolic presence and greater concentration in the liver as compared with other tissues. The interpretation and correlation of ALT levels with liver injury also appears to be clinically important [4]. A significant hepatic steatosis was found only in ADH- vs. ADH+ deer mice fed 3.5% EtOH using H & E staining (Fig 1A, f), which correlates well with increased ALT levels.

In the liver, ethanol is predominantly metabolized by class 1 ADH (a classical oxidative pathway), which has low Km (high affinity) for ethanol among all members of ADH family. Studies from our lab and others have shown that hepatic ADH activity is impaired under chronic alcohol condition [5-7]. Of importance, rodents are known to metabolize ethanol at much higher rate compared to that in humans [8]. Since ADH- deer mice, a natural variant of Peromyscus maniculatus sp mimics the condition of chronic alcoholics, we used ADH- deer mouse model to address the questions regarding mechanisms and metabolic basis of alcoholic liver disease.

Refernces

[1] E.C. Jensen, Quantitative analysis of histological staining and fluorescence using ImageJ, Anatomical record, 296 (2013) 378-381.

[2] J. Han, R.J. Kaufman, The role of ER stress in lipid metabolism and lipotoxicity, J Lipid Res, 57 (2016) 1329-1338.

[3] M.C. Kew, Serum aminotransferase concentration as evidence of hepatocellular damage, Lancet, 355 (2000) 591-592.

[4] W.R. Kim, S.L. Flamm, A.M. Di Bisceglie, H.C. Bodenheimer, D. Public Policy Committee of the American Association for the Study of Liver, Serum activity of alanine aminotransferase (ALT) as an indicator of health and disease, Hepatology, 47 (2008) 1363-1370.

[5] B.S. Kaphalia, B.I. Ghanayem, G.A. Ansari, Nonoxidative metabolism of 2-butoxyethanol via fatty acid conjugation in Fischer 344 rats, J Toxicol Environ Health, 49 (1996) 463-479.

[6] J. Panes, J. Caballeria, R. Guitart, A. Pares, X. Soler, M. Rodamilans, M. Navasa, X. Pares, J. Bosch, J. Rodes, Determinants of ethanol and acetaldehyde metabolism in chronic alcoholics, Alcohol Clin Exp Res, 17 (1993) 48-53.

[7] H. Nuutinen, K.O. Lindros, M. Salaspuro, Determinants of blood acetaldehyde level during ethanol oxidation in chronic alcoholics, Alcohol Clin Exp Res, 7 (1983) 163-168.

[8] D.J. Livy, S.E. Parnell, J.R. West, Blood ethanol concentration profiles: a comparison between rats and mice, Alcohol, 29 (2003) 165-171.

Reviewer 2 Report

This report provides substantial data to investigate how alcohol dehydrogenase presence or not, AMPK, downstream signals and alcohol intake led to steatosis and liver injury. Here are my comments:

1) About the Methods, the authors should describe more about the way of feeding the drinking diet containing EtOH to the mice. Was it force feeding? How do the authors ensure that the different groups (0, 1, 2 and 3.5%) had the same or similar volume intake? Did the mice show preference for non-alcoholic drink or 3.5%?

2) Fig 1 data need quantification. Was GPR78 increase really significant in the IHC expt? Its increase was not significant in the western blot expt. Therefore I am totally not convinced that ER stress took place, in addition, because other ER stress markers were not increased.

3) There are a lot of western blot data not shown. I suggest the authors could provide a Table summarizing the changes (increase/decrease/no change) of the protein levels of all the signals investigated.

4) To help the readers, the authors are suggested to create a summarizing cartoon diagram to illustrate the proposed pathway leading to steatosis and liver injury after EtOH intake.

Author Response

Reviewer #2

This report provides substantial data to investigate how alcohol dehydrogenase presence or not, AMPK, downstream signals and alcohol intake led to steatosis and liver injury. Here are my comments:

 Comment 1:  About the Methods, the authors should describe more about the way of feeding the drinking diet containing EtOH to the mice. Was it force feeding? How do the authors ensure that the different groups (0, 1, 2 and 3.5%) had the same or similar volume intake? Did the mice show preference for non-alcoholic drink or 3.5%?

 Response: In this study, we used nutritionally balanced Lieber-DeCarli liquid diet for chronic ethanol feeding or isocaloric control liquid diet where ethanol derived calories were replaced by maltose dextrin for pair feeding [1,2].  Therefore, mice were provided with nutritionally balanced liquid diets as their sole source of food and water, where they consume liquid diet voluntarily [3], thus it was not necessary to force feed the mice with either ethanol or isocaloric control liquid diet. Further, Peromyscus maniculatus sp (Deer Mice) are shown to have high self-selection for alcohol [4]. Both, ethanol-fed and pair-fed controls consumed approximately same number of calories. The calories derived from 1 mL liquid diets are equal to1 kcal [5]. A maximum tolerable dose of ethanol in deer mice was found to be 3.5%, and beyond this dose we observed an increased mortality in these mice [6].

 Comment 2: Fig 1 data need quantification. Was GPR78 increase really significant in the IHC expt? Its increase was not significant in the western blot expt. Therefore, I am totally not convinced that ER stress took place, in addition, because other ER stress markers were not increased.

 Response: Please see the earlier response for comment by reviewer 1(see attached).

Comment 3: There are a lot of western blot data not shown. I suggest the authors could provide a T able summarizing the changes (increase/decrease/no change) of the protein levels of all the signals investigated

Response: As pointed out by the reviewer, we have now provided tables which summarizes the changes in protein levels by Western blot analysis conducted in this study. 

Comment 4: To help the readers, the authors are suggested to create a summarizing cartoon diagram to illustrate the proposed pathway leading to steatosis and liver injury after EtOH intake.

Response: We have now included a flow chart to demonstrate the prosed pathway for pathogenesis of ALD.

Refernces

[1] H. Fernando, K.K. Bhopale, P.J. Boor, G.A. Ansari, B.S. Kaphalia, Hepatic lipid profiling of deer mice fed ethanol using (1)H and (3)(1)P NMR spectroscopy: a dose-dependent subchronic study, Toxicology and applied pharmacology, 264 (2012) 361-369.

[2] B.S. Kaphalia, K.K. Bhopale, S. Kondraganti, H. Wu, P.J. Boor, G.A. Ansari, Pancreatic injury in hepatic alcohol dehydrogenase-deficient deer mice after subchronic exposure to ethanol, Toxicology and applied pharmacology, 246 (2010) 154-162.

[3] G.E. McClearn, Animal models in alcohol research, Alcohol Clin Exp Res, 12 (1988) 573-576.

[4] W. Poley, L. Mos, Emotionality and alcohol selection in deer mice (Peromyscus maniculatus), Q J Stud Alcohol, 35 (1974) 59-65.

[5] F. Guo, K. Zheng, R. Benede-Ubieto, F.J. Cubero, Y.A. Nevzorova, The Lieber-DeCarli Diet-A Flagship Model for Experimental Alcoholic Liver Disease, Alcohol Clin Exp Res, 42 (2018) 1828-1840.

[6] K.K. Bhopale, H. Wu, P.J. Boor, V.L. Popov, G.A. Ansari, B.S. Kaphalia, Metabolic basis of ethanol-induced hepatic and pancreatic injury in hepatic alcohol dehydrogenase deficient deer mice, Alcohol, 39 (2006) 179-188.

Round 2

Reviewer 1 Report

The quantification of GRP78 and the addition of electron micrographs (EM) of the liver sections showing dilation and swelling of the endoplasmic reticulum (ER) and ER cisternae only in ADH- vs. ADH+ deer mice fed  3.5% EtOH significantly improved the manuscript. I would add that the EM also shows megamitochondria in ADH- mice which is recognized as a hallmark of alcoholic liver diseases [Am J Pathol. 2019 Mar; 189(3): 580–589Liver. 1984 Feb;4(1):29-38)].

Reviewer 2 Report

Revision is OK.